# Evaluating the long-term operational performance of a large-scale inland terminal: A discrete event simulation-based modeling approach

Punyaanek Srisurin[1], Phipat Pimpanit[2], Pisit Jarumaneeroj[2,3] *

**1** Department of Civil and Infrastructure Engineering, Asian Institute of Technology, Pathum Thani, Thailand, **2** Department of Industrial Engineering, Chulalongkorn University, Bangkok, Thailand, **3** Regional Centre for Manufacturing Systems Engineering, Chulalongkorn University, Bangkok, Thailand

☯ These authors contributed equally to this work.

\* pisit.ja@chula.ac.th

**Data Availability Statement:** The data for this study, as well as the simulation models, have been provided in the link below: https://github.com/ORCChula/LICD.

## Abstract

Inland terminals, or dry ports, have played an important role in multimodal transportation networks as transportation hubs that provide connections between seaports and hinterland economies. While important, evaluating the operational performance of a dry port is especially challenging since it depends not only on internal factors, such as the variety and number of container handling equipment (CHE) deployed, but also on other external factors, including changes in transportation policies and container demands experienced by a dry port. To properly evaluate the holistic performance of a dry port while considering all the aforementioned factors, a discrete event simulation (DES) framework is herein developed and applied to the Ladkrabang Inland Container Depot (LICD)—one of the largest dry ports in Southeast Asia—under various operational settings. Despite complicated internal operations, the devised DES framework has shown itself useful in the analyses of LICD, due largely to its flexibility that allows users to include sophisticated operational rules into models. According to our computational results, the current LICD operation is markedly ineffective as the usage rates of all CHE types are relatively low and varied across gate operators —especially the yard truck whose values range between 2.46% and 11.15% on yearly average. We also find that, by redesigning the LICD and its internal operations, the LICD's performance could be substantially enhanced—even with fewer numbers of CHE. Regarding the four CHE types, the reach stacker seems to limit LICD's capability, as its utilization tends to first reach the maximum allowable rate of 75%, while the rubber tyred gantry crane could help boost the usage rate of yard trucks, which, in turn, results in reduced container dwelling times. Nonetheless, the modified LICD could accommodate up to 140% of the current container demand before it experiences operational difficulties induced by the saturation of container flow from rail transportation.

**Funding:** PJ is supported by the EU Horizon 2020 Marie Sklodowska-Curie Research and Innovation Staff Exchange project GOLF (reference number GOLF-777742). The funder had no role in study design, data collection and analysis, decision to publish, or preparation of the manuscript.

**Competing interests:** The authors have declared that no competing interests exist.

## 1. Introduction

While liner shipping companies are the main contributors to international trade, a vast amount of goods traded could not be, however, transported directly from manufacturers to customers through a single transportation mode. Generally, globally traded goods are channeled through multiple routes via multiple modes of transportation that require a number of transshipments at various transportation hubs [1]. A seaport, or a coastal terminal, is a prominent example of such hubs that allows goods to be transshipped from cargo vessels to trains or trucks, and vice versa, for hinterland access. Besides coastal terminals, there are also inland terminals—typically referred to as dry ports [2, 3]—that help expand the coverage of water transportation while providing economies of scale to shipping companies [4].

According to the United Nations Conference on Trade and Development (UNCTAD), a dry port is defined as "a common user facility with public authority status, equipped with fixed installations, offering services for handling and hinterland of seaports". Technically speaking, a dry port is a type of transportation hubs that helps facilitate hinterland transportation, including depositing, transferring, assembling, and controlling cargo flow [5, 6]. Dry ports can also help reduce waiting time due to consular formalities at seaports, which, in turn, allows gate operators to load cargo as soon as vessels are ready [7].

Based on studies by [8–10], the impacts of these facilities—especially dry ports—on the efficiency of multimodal transportation have become increasingly evident due to globalization and intensified trading. Nonetheless, research related to dry ports typically focuses on strategic decisions, for example determining the most cost-effective locations of dry ports [11–13] or assessing the strategic roles of dry ports in multimodal transportation networks from various players' perspectives [14–18], without taking into consideration other important operational factors, including changes in a dry port's infrastructure—and therefore its performance—as well as those of multimodal transportation environments over time [19].

We also find that research related to dry port performance evaluation is rather limited, especially at the holistic level, where all container handling equipment (CHE) across internal facilities is integrated into one single framework. Asgari et al. [20], for instance, adopted the AHP method to evaluate the performance of major UK ports based only on the opinions of port managers and experts in the logistical domain. Lin et al. [21], on the other hand, used Arena software to develop a simulation-based model for dry ports to minimize CHE investment while improving the dynamism of CHE at operational levels. Discrete event simulation (DES) models mimicking operations within container terminals were also proposed by [22, 23], but with different foci and performance measures. Lastly, Tang et al. [24] have developed a simulation-based model using AnyLogic platform to improve the power peak of CHE under different operational policies.

It could be seen from the previous research that simulation has played an important role in the evaluation of inland/coastal terminal performance [25–28], as well as that of other logistics-related facilities [29, 30], whose detailed settings may vary depending on the investigated operations and platforms. However, most studies discussed so far have focused only on one or part of inland/coastal terminal operations with a fixed set of CHE. This may not properly reflect the true performance of inland/coastal terminals as many CHE types are actually shared resources that could be assigned to different operations across terminal areas. Top loaders and reach stackers, for instance, are both crane-based equipment typically deployed for container relocation between vehicles (external/internal trucks and trains) and container yards. Yet, top loaders can relocate only empty containers, while reach stackers can perform relocation activities for all container types. Since reach stackers are more flexible than top loaders, the former tend to be utilized in various operations across terminals rather than being used in any specific part of a terminal as is the case with the latter.

In light of this gap, this paper aims to build a more realistic DES model that takes into account not only the operations of a dry port with shared CHE but also other managerial factors that could potentially affect a dry port's long-term performance, including infrastructure, multimodal transportation policies, and the future course of a dry port from the regulator's point of view. For ease of discussion, the proposed DES model will be developed based on the Ladkrabang Inland Container Depot (LICD)—the first and only containerized dry port in Thailand—and its development plan through the help of SIMIO simulation program. With this proposed model, related players should be able to accurately evaluate the current dry port's performance, as well as that of a new layout at various development stages prior to the implementation. The proposed model will also allow the players to conduct detailed analyses under different scenarios, which will, in turn, help enhance the concurrent utilization of expensive CHE, the dry port, and the respective multimodal transportation network. It is worth remarking that, while the devised DES model has been specifically constructed for the analyses of LICD, such a modeling framework could be extended to other dry ports or similar logistics-related facilities with the same or different CHE types; but this may require modifications on the model's (sub)modules, as well as those on the operational rules of investigated facilities.

The remainder of this paper is organized as follows. Section 2 provides detailed information of the LICD, together with its operations under both the current and the proposed LICD layouts. DES models representing the LICD are then introduced in Section 3, followed by comprehensive simulation results in Section 4. Lastly, Section 5 summarizes the present work and discusses some future research directions.

## 2. Ladkrabang Inland Container Depot (LICD)

### 2.1. An overview of the LICD

Ladkrabang Inland Container Depot (LICD) is the first and only containerized dry port in Thailand. According to the State Railway of Thailand (SRT), the LICD has been receiving an increasing amount of cargo volume each year. The quantity of twenty-foot equivalent units (TEUs) handled at the LICD has seen a significant growth of over 200% since its establishment in 1996—the number in 2020 recently topped 1,260,054 TEUs. The two key drivers that help boost container flow at the LICD are its strategic location and infrastructure. In terms of location, the LICD is located close to the majority of Thai suppliers and the two main seaports, namely the ports of Bangkok and Laem Chabang, with a combined annual capacity of over 8.7 million TEUs—about 1 million TEUs go through the port of Bangkok and the remaining are handled at the port of Laem Chabang. Furthermore, the LICD is equipped with rail connections not only to these seaports but also to a number of regional industrial estates, which further help connect them with a water gateway to international trade.

Due to the rapid growth of container flow at the LICD, the Port Authority of Thailand (PAT), with support from the Thai government, has thence decided to redesign the LICD so that it could serve as a transportation hub with improved connections to other inland terminals and regional seaports in Southeast Asia. Although the LICD is currently undergoing a major transformation, at both strategic and operational levels, according to the PAT's multiphase development plan, it is somewhat difficult to evaluate the current performance of LICD, as its present operation involves several gate operators performing different handling operations within the same site and with the same infrastructure. Particularly, there are currently six gate operators, each of whom independently operates one LICD gate (or a terminal) using the same railway tracks located in the middle of LICD. Each gate operator also owns private handling equipment that is utilized only within the operator's service area, which further

complicates analysis and comparison between the current and the future LCID layouts, as depicted in Fig 1.

Although the LICD operation is complicated, we show that such an operation could be transformed into a DES model similar to those described by [31, 32], where states, and thus the LICD's performance, depend on sequences of discrete events at specific time periods [33]. Nevertheless, we do need a proper modeling formalism capable of capturing the dynamics of CHE, as well as those of other entities operating in the LICD. Among numerous commercial simulation platforms that are available, we have found that SIMIO has superior advantages as it allows users to visualize a model in a three-dimensional view [34]. It is also

(a)

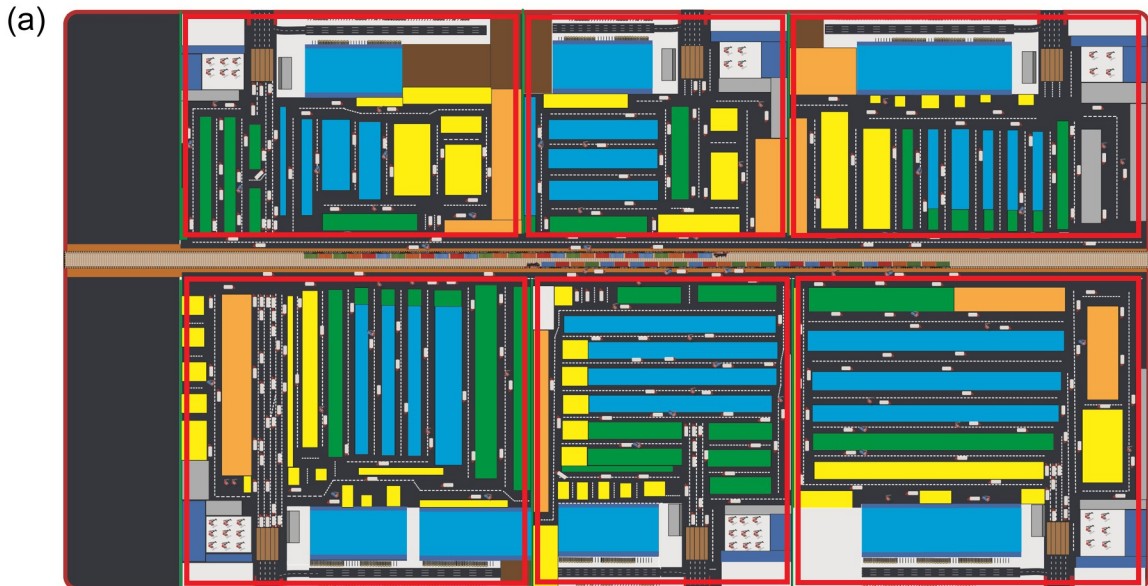

(b)

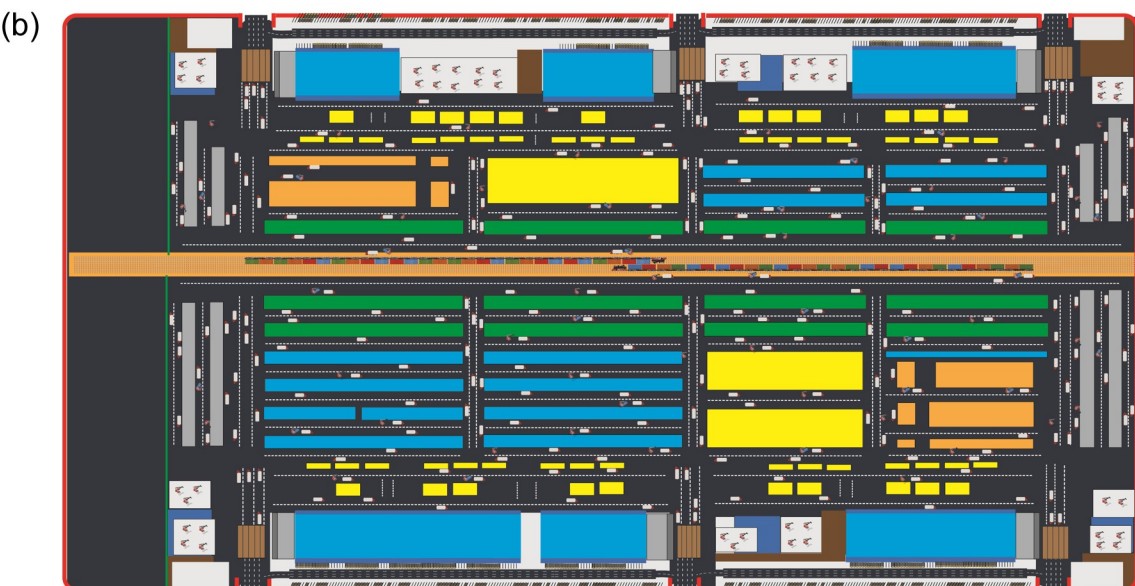

**Fig 1. Layouts of the current and the projected LICD by PAT.** (a) The current LICD layout with six independent gate operators. (b) The new LICD layout.

flexible, especially in facilitating the control of shared resources in a complicated operational environment, as users are enabled to include sophisticated operational rules into the model.

With help of SIMIO, we can successfully construct a DES model, referred to as a base model, representing the current LICD operation, as well as those of PAT's projections at different development stages. We also show that, with these models, cost-effective CHE planning could be devised and executed at proper decision epochs.

## 2.2. LICD layouts

According to the LICD's current operation, each gate operator generally performs tasks related to storing, importing, and exporting four freight types—namely, full container loads (FCLs), less than container loads (LCLs), refrigerated containers (reefers), and empty containers—each of which requires different kinds of CHE owned by relocation operators. Within each operator's designated area (see Fig 1a.), there are also six sub-zones with specific functions. These sub-zones are;

1. cargo consolidation and deconsolidation depot (warehouse), where goods are packed into loads of containers for export and import,

2. container maintenance and repairing depot, where empty containers are inspected and repaired before being transferred to other internal facilities, such as the cargo consolidation and deconsolidation depot or empty container yards,

3. export container yard, where export containers are stored for outbound transportation by trucks and trains,

4. import container yard, where import containers are stored before being towed for further operations,

5. empty container yard, where empty containers are stacked and relocated within and outside the LICD, and

6. refrigerated (reefer) container yard, where temperature-controlled containers requiring electrical socket connections are stored.

In order to move container freight within and across the aforementioned areas, the following CHE types are typically deployed.

- Top loader (TL): a TL is a type of cranes capable of stacking and unstacking empty containers (3.75 t) in yard and depot areas. It is also used to transfer empty containers from/to trucks for further inland transportation.

- Reach stacker (RS): an RS is another kind of cranes that performs similar tasks to those performed by a TL. However, it is equipped with strong hoists capable of lifting both empty and fully loaded containers (29 t).

- Yard truck/internal truck (YT): a YT is an internal vehicle used to relocate containers within and across the LICD.

- Rubber tyred gantry crane (RTG): an RTG is a crane, normally positioned over railway tracks, capable of vertically stacking and unstacking containers from/to trains and trucks. It is worth noting that, currently, the LICD has no RTGs due to its fragmented operations; but this CHE type will be introduced in the new LICD layout in order to expedite operations within rail areas.

Instead of having six independent gate operators, the new LICD will be operated by a single operator, whose layout will comprise two expansive terminals and four railway tracks, as shown in Fig 1b. Under this new setting, an RTG will be introduced and used along the railway tracks for both import and export operations. Nonetheless, the RTG will not be procured at the early development stage due to its relatively high investment cost. Rather, it will be procured when container demands increase during the latter phases.

## 2.3. LICD operations

In terms of operations, Fig 2 summarizes handling operations for both import and export containers at the LICD. Regarding import containers, once they arrive (either by trains or trucks), they will be unloaded and relocated to the designated areas by CHE. The carriers may then leave the LICD or be assigned to other handling operations (if allowed). Export containers, on the other hand, will be moved from yards to trains or trucks, according to their schedules, by

**Fig 2. LICD operations for both import and export containers.**

the same or different sets of CHE depending on a gate operator's policies and the CHE's functions. For instance, gate operators may use RSs to relocate export containers from yards to YTs destined to the rail zone, or they may allow RSs to directly load export containers from yards onto trains without the help of YTs so as to avoid double handling (this operation in practice is sometimes referred to as a shifting operation).

Although the flows of import and export containers remain the same for the current and the new LICD layouts, container handling operations in the rail zone of the new LICD are significantly different from those of the current one, mainly because of the RTG. To be precise, loading and unloading from trains in the new LICD may be performed by both RSs and the RTG, while these operations may only be performed by RSs in the current layout. Furthermore, as the RTG is capable of moving containers among four railway tracks, while RSs can only move containers from/to railway tracks adjacent to terminals, the movement of both CHE and containers in the new layout is more dynamic.

To properly track containers and CHE in both layouts, we have adopted a simple rule-based strategy similar to [35–37] for both CHE assignment and its respective routing, where containers are assigned to the first closest available CHE; and, if no such CHE is available, the containers will be stalled until the required CHE is ready. Once assigned, the routing of CHE will be subsequently initiated. In this regard, it could be seen that CHE routing in the current layout is quite straightforward as the CHE is privately owned and must be routed within a small designated area. However, CHE routing in the new layout is relatively more complicated and dispersed as its respective service area is significantly expanded.

## 2.4. Changes in transportation environments

In addition to layout, rail transportation policy is found to be another factor that might affect the long-term efficiency of dry ports [27, 38]. As such, the LICD will be explored under two different transportation environments, in which transportation ratios between road and rail have been set at 80:20 (base ratio) and 50:50, respectively.

The first scenario is derived from the fact that almost all the containers handled at the LICD are currently transported by trucks (about 80% of total container flow)—despite the fact that there is currently a railway connection that connects the LICD and the two main seaports. The reason for this is the SRT's management issues that frequently result in transportation delays. Nonetheless, the SRT has proposed a new transport contingency plan to the PAT that aims to improve its timetables, as well as the efficiency of the rail transportation system, so that the desired transportation ratio of 50:50 between road and rail can be achieved.

Finally, as the container flow at the LICD is expected to increase according to the PAT's projection, several CHE plans must also be assessed so that we can maintain a high service level while properly utilizing expensive CHE.

## 3. Methodology

### 3.1. DES assumptions

As has been illustrated in the previous section, the LICD operations are rather complex, especially in the new layout with the rubber tyred gantry crane (RTG). We therefore need proper mechanisms and some assumptions to help simplify the systems into tractable DES models.

In terms of structure, our DES models could be divided into four main modules, each of which may involve one or more internal operations, as follows.

- Road and rail infrastructure: this module contains transportation systems within the LICD. It is also used to control the movement of containers/CHE based on LICD's road and rail networks.

- CHE movement: this module helps control the motion of CHE according to handling tasks.

- Container movement: this module controls the sequences of both inflow and outflow containers.

- Maintenance and repair: unlike the previous modules, this module is a supportive module that specifically controls the movement of empty containers requiring maintenance and repair.

As with other DES models, when containers arrive at or depart from the LICD, the model's states will change in accordance with predefined rules and patterns of arriving/departing containers. In this regard, we may categorize DES input into two groups, namely (*i*) the arrival/ departure of containers by trucks and (*ii*) the arrival/departure of containers by trains.

For simplicity, we assume that all containers handled at the LICD are 40-foot long. More-over, all types of containers may be transported by trucks, but only full container loads could be loaded onto trains. Finally, the numbers of both inbound and outbound containers assigned to each gate operator in the current layout are based solely on the operator's market share due to lack of operator-level information.

## 3.2. DES models

Our proposed DES models are built into the SIMIO simulation platform, where all LICD operations are categorized and controlled through four different operation systems, namely (*i*) terminal operation system, (*ii*) container operation system, (*iii*) CHE operation system, and (*iv*) warehouse operation system. To give a proper visualization, Fig 3 shows an overview structure of our proposed DES models. In the figure, it could be seen that every operation within the LICD is performed by CHE. A proper tracking mechanism is therefore crucial for each system, and this mechanism may be regarded as one of the key variables in the DES models.

**3.2.1. Terminal operation system.** The terminal operation system could be regarded as a supervisory controller responsible for all operations taking place within the LICD. It is also used to initialize DES input, including LICD's functional areas, road and rail networks, work scopes of entities, and interactions among internal/external entities. This system, however, does not include operations outside the LICD, such as the arrival and departure of external trucks, as they are handled by the container operation system.

**3.2.2. Container operation system.** The container operation system helps control the arrival and departure of containers via road and rail networks. Particularly, when containers arrive at the LICD by external trucks, they will be assessed whether they need consolidation or deconsolidation. If consolidation or deconsolidation is not required, the containers will be assigned and routed to respective container yards; else they must first visit a cargo consolidation and deconsolidation depot for stuffing/unstuffing operations before being moved to designated container yards. In case the arriving containers are empty, they must visit the container maintenance and repairing depot for cleaning and repairing before being transferred to empty container yards. Once unloading tasks have been done, external trucks may leave the LICD or be assigned to other handling tasks.

For clarity, Fig 4 shows the detailed process for arriving and departing containers with no consolidation/deconsolidation via road networks, where external trucks are allowed to visit

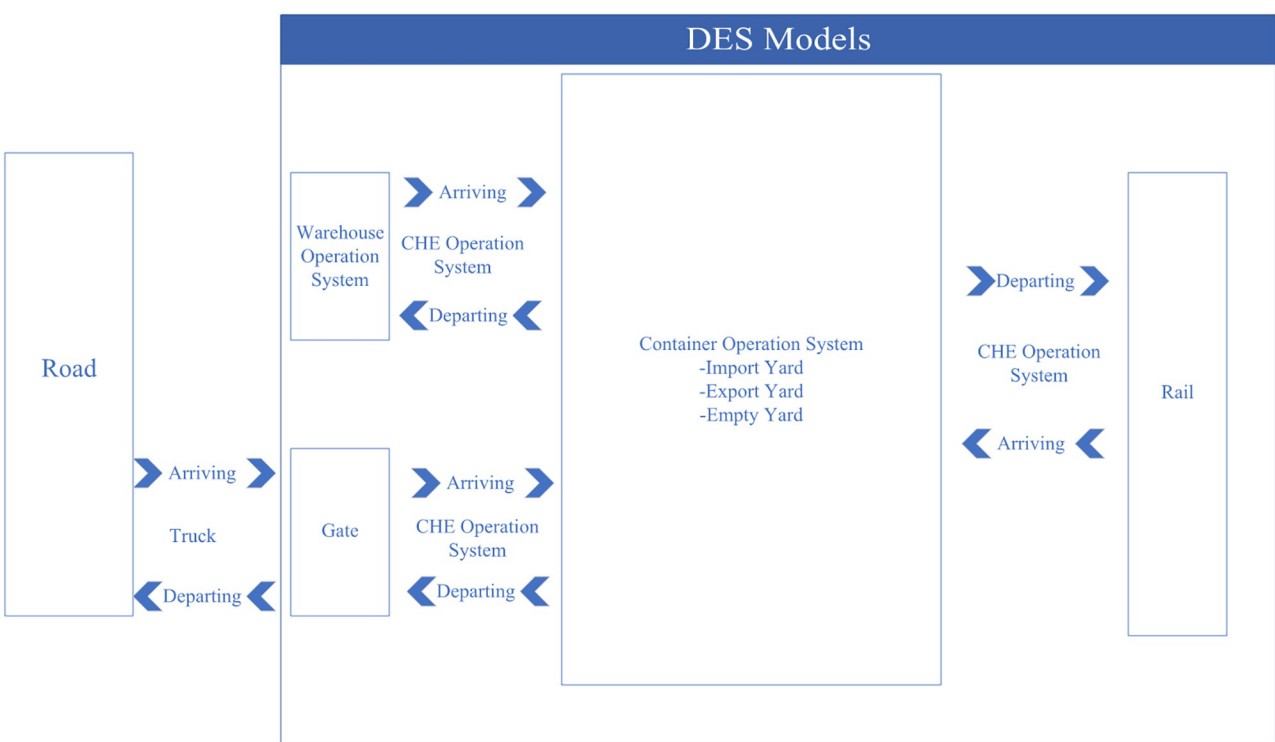

**Fig 3. An overview of the proposed DES models.**

the LICD for the delivery of both import and export containers. The detailed process for containers requiring consolidation/deconsolidation is more complicated, as it involves the use of internal equipment to either move empty containers to the container maintenance and repairing depot or to move full container loads to export container yards.

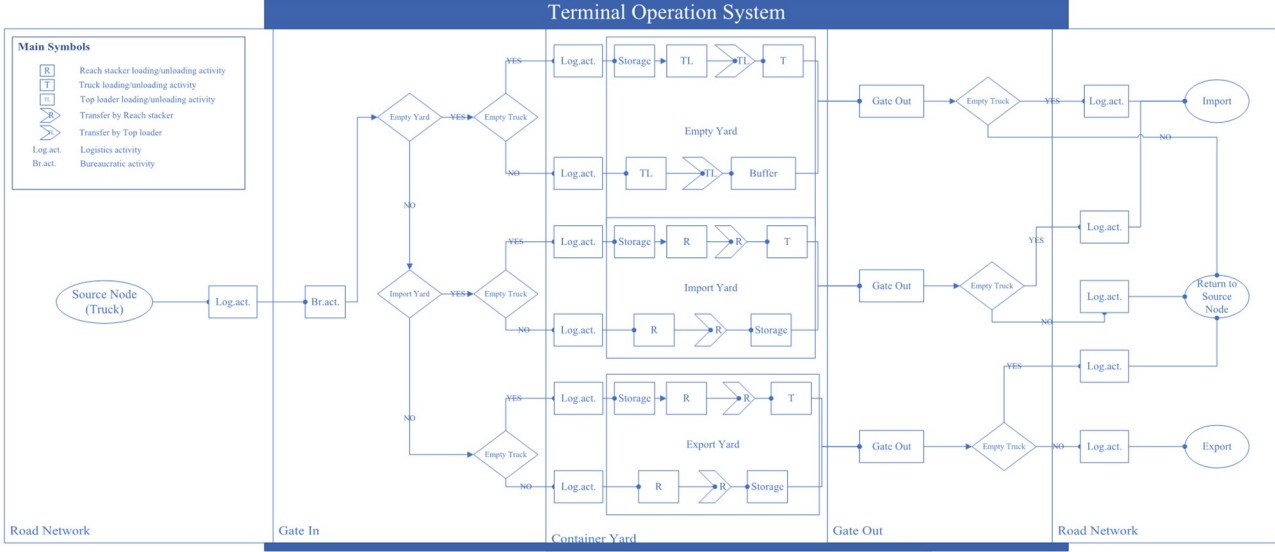

**Fig 4. The detailed process for arriving and departing containers with no consolidation/ deconsolidation via road networks.**

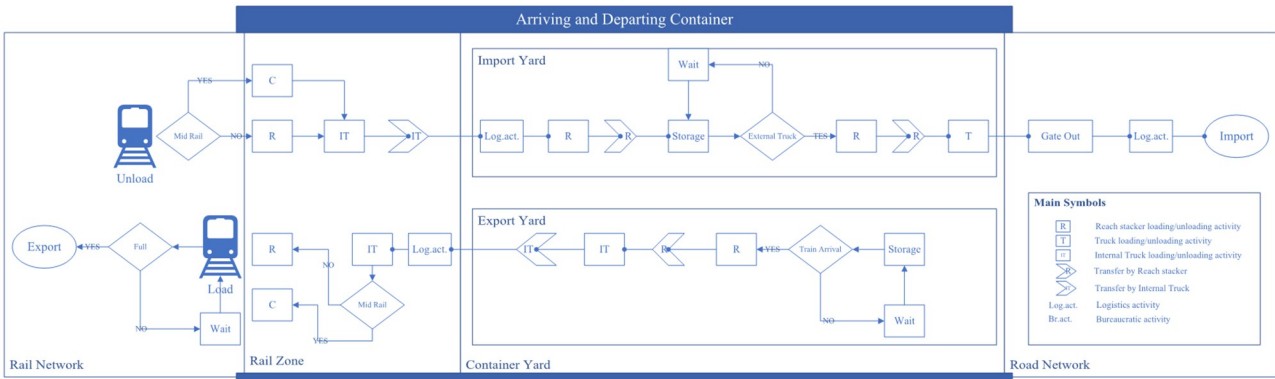

**Fig 5. Operational flow of containers arriving and departing the LICD by trains.**

Unlike the arrival and departure of containers by trucks, containers arriving and departing by trains either originate from or destined to container ports (mostly the port of Laem Chabang). Furthermore, each train trip can accommodate up to 64 TEUs due to limited train capacity.

When a cargo train arrives at the LICD, arriving containers will be unloaded onto internal trucks—or yard trucks—and then taken to designated container yards until the cargo train is emptied. Once all import containers are unloaded, export containers will be later loaded onto the train with help of yard trucks and other CHE types in rail areas. Nonetheless, containers located at yards close to railway tracks may be loaded onto the train without the use of yard trucks via shifting operations. Once the cargo train is loaded, it may leave the LICD with its export cargo. To better visualize this process, Fig 5 below shows the operational flow of containers arriving and departing the LICD by trains.

**3.2.3. CHE operation system.** The CHE operation system involves CHE assignment and its respective routing within the LICD, as illustrated in Fig 6. From the figure, it could be seen

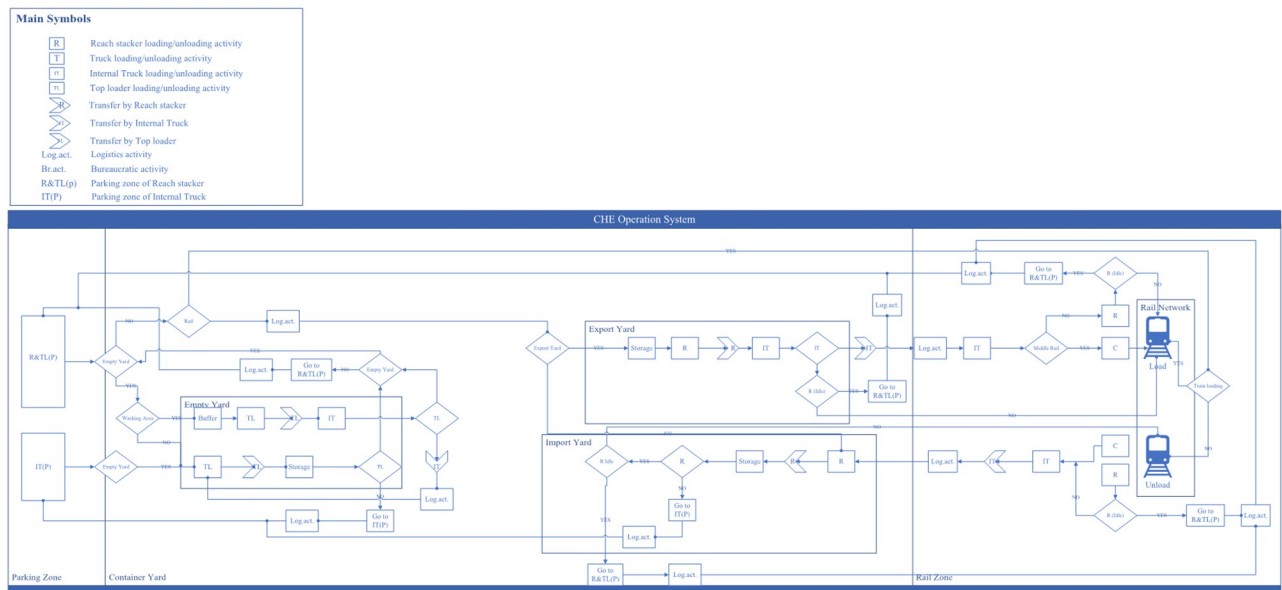

**Fig 6. Operational flow of the CHE operation system.**

that all CHE-related operations begin in CHE parking areas, where idle CHE is placed. Each CHE parking area comprises two sub-zones: one for idle cranes (reach stackers and top loaders) located inside the container yards close to gate and warehouse areas, and another for idle internal trucks located in front of warehouses.

Recall that, while there are four different types of CHE deployed in the LICD—namely, reach stacker (RS), top loader (TL), rubber tyred gantry crane (RTG), and yard truck (YT)— the RTG will only be introduced in the new layout. Furthermore, some types of CHE can perform specific handling functions only within specific zones. RSs and TLs, for example, are generally dispatched for container handling operations in three different zones—warehouses, container yards, and rail areas—while RTGs with the same container handling functions can be operated only on railway tracks.

Based on these facts, the CHE operation system must be properly designed to reflect the movement of CHE and containers by taking into consideration the CHE's intended functions and its respective movements in different operational areas. More formally, when containers arrive at warehouse areas for consolidation/deconsolidation via road networks, RSs must be first assigned to help move such containers from external trucks to the warehouses. If the arriving containers are imported, the goods will be unstuffed and empty containers will be loaded onto yard trucks heading to the container maintenance and repairing depot for further operations by TLs (pairs of top loaders and yard trucks will be used once again for the relocation of empty containers from the container maintenance and repairing depot to empty container yards). Otherwise, the goods will be restuffed and the resulting full container loads will be loaded onto yard trucks by RSs and sent to export container yards.

When yard trucks arrive at container yards for either container storage or relocation, another set of cranes at yards will be dispatched to help complete the operations; but this depends on the type of arriving loads as TLs can only be assigned to empty container handling tasks.

Unlike warehouse and container yard areas, CHE operations within rail areas are somewhat different for the current and the new LICD layouts, due largely to the introduction of an RTG as shown in Figs 7 and 8. Initially, when a cargo train arrives at the LICD, all external trucks must be cleared from terminals to avoid complications from consular formalities. Then, a set of RSs and yard trucks will be assigned to rail areas to unload and relocate import containers to their respective yards until the cargo train is empty. Once the train is ready for export, export containers will be moved from export container yards to rail areas and then lifted onto the train by RSs and yard trucks. However, with the presence of an RTG in the new LICD layout, container loading and unloading in the rail areas could be expedited, as loading and unloading could be performed by both RSs and the RTG.

**3.2.4. Warehouse operation system.** The warehouse operation system is the last system specifically developed for consolidation and deconsolidation activities at warehouses, in which loads from external trucks are unstuffed, stored, and restuffed back to containers for export. As with previous systems, once these operations have been performed, related CHE will be dispatched depending on the type of containers handled. For example, top loaders will be dispatched for empty container handling tasks, whereas reach stackers will be deployed for tasks related to full container loads. For ease of discussion, the operational flow of warehouse operation system is illustrated in Fig 9.

# 4. Simulation results

## 4.1. Parameter and scenario settings

We have run the DES models based on the operations of LICD between October 2018 and September 2019. Information concerning these operations was provided by the State Railway

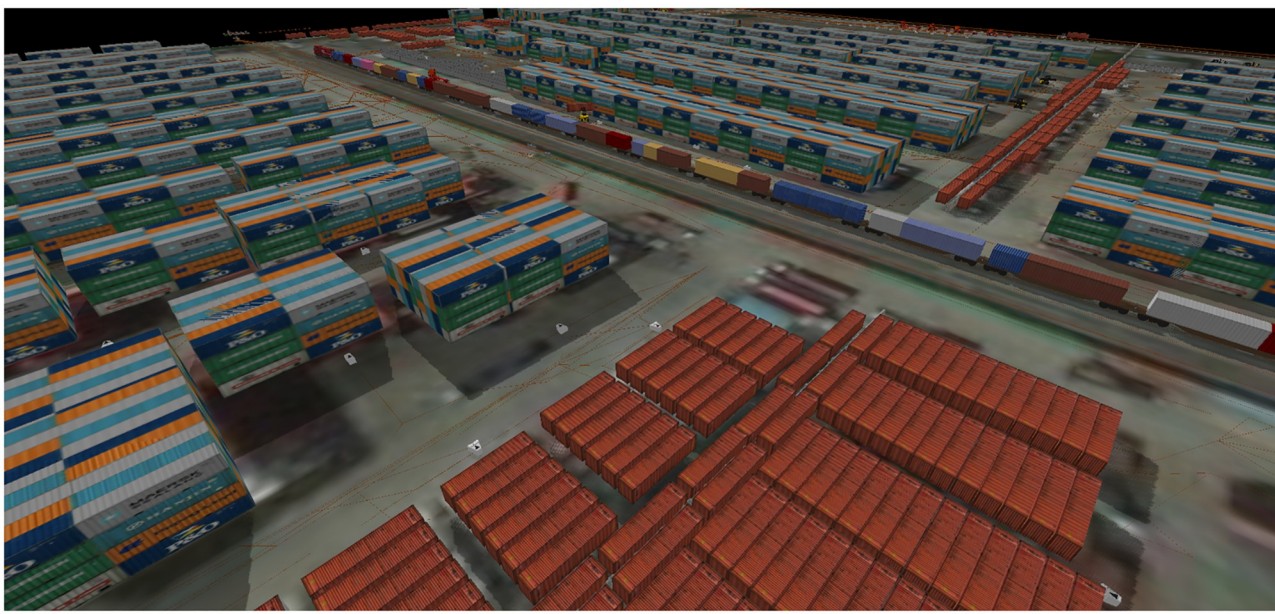

**Fig 7. Rail area of a DES model representing the current LICD layout with no RTGs.**

of Thailand (SRT) and the Eastern Sea Laem Chabang Terminal Co., Ltd. (ESCO). The main LICD parameters extracted from these data included processing time for each handling activity, together with its respective fitted distribution, and CHE speeds, as summarized in Table 1. In addition to these basic parameters, information concerning monthly numbers of containers handled at the LICD was also provided, although it lacked details at the daily operational level which, in turn, required further assumptions about the arrival of each container type. For simplicity, we have assumed that the probability distribution of arriving containers follows a uniform distribution, but with different mean values and standard deviations over the year.

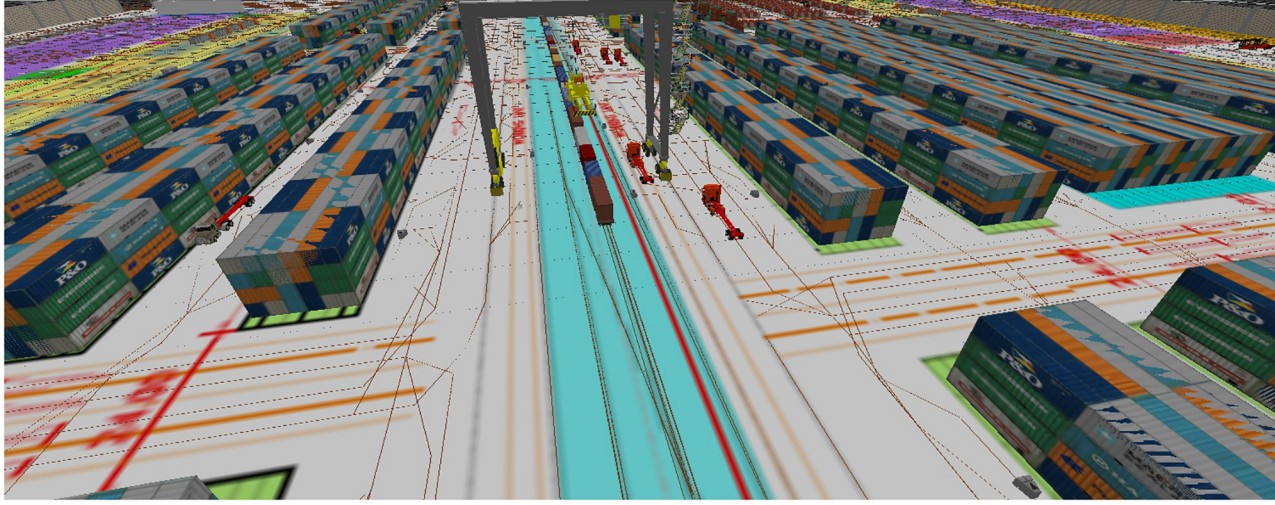

**Fig 8. Rail area of a DES model representing the new LICD layout with an RTG.**

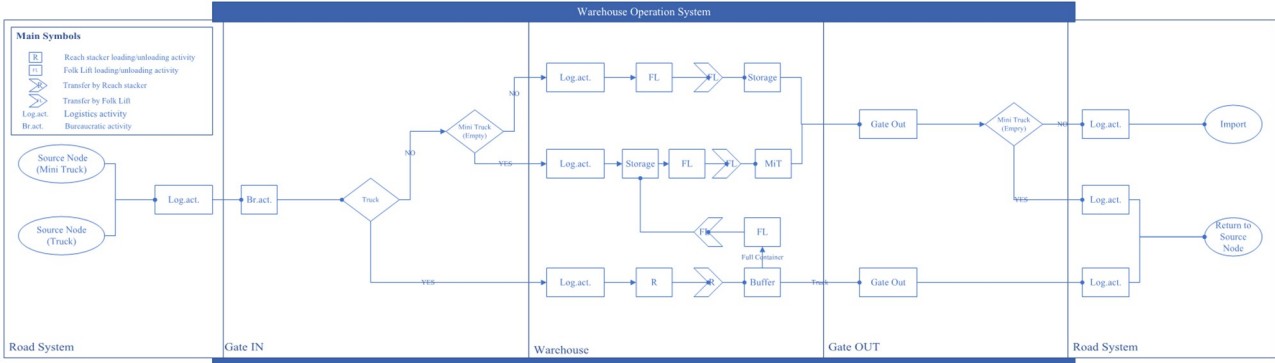

**Fig 9. Operational flow of the warehouse operation system.**

In terms of setting, the LICD will be evaluated based on six different scenarios, where scenarios S1 and S2 involve the current LICD operation under two different transportation policies, imposed by the Port Authority of Thailand (PAT), while scenarios S3–S5 involve that of the new LICD layout. Each scenario has different CHE numbers that align with the PAT's multi-phase development plan. Scenarios S1 and S2, on the one hand, and scenarios S3–S5, on the other hand, are summarized in Tables 2 and 3, respectively. Lastly, scenario S6 concerns the sustainability of the new LICD when container demand has increased incrementally.

Standard CHE usage rates are set based on the PAT's experience, as shown in Table 4, where the minimum and maximum numbers of moves for each CHE type are listed. Based on Table 4, the acceptable FCL crane (RS) usage, for instance, ranges between 75 (35% utilization) and 150 moves per day (75% utilization), while that of empty containers (TL) ranges between 40 (20% utilization) and 150 moves daily (75% utilization).

It is worth remarking that, while high CHE utilization is preferable in terms of investment, excessive CHE movements might lead to high operational costs and delays—especially during

**Table 1. Main parameters extracted from the information provided by SRT and ESCO.**

| Parameter | Value and Fitted Distribution | Unit |
|---|---|---|
| Container transferring time from trucks to yards | UNIF(2,2.5) | min/box |
| Container transferring time between yards | TRIA(2,3,7)*UNIF(2,2.5) | min/box |
| Container transferring time by an RTG in the rail zone | UNIF(2,4,3) | min/box |
| Washing and repairing time | POIS(32) | min/box |
| Waiting time for empty containers before being transferred to other operations | POIS(8) | h/box |
| Stuffing and unstuffing time | POIS(15) | min/box |
| RS and TL speed | 15 | km/h |
| RTG speed | 7.8 | km/h |
| YT speed | 30 | km/h |
| Driver shift | 8 | h/shift |

**Table 2. Settings for scenarios S1 and S2 (the current LICD layout).**

| | | | Current Layout | | | | | | | | | | |
|---|---|---|---|---|---|---|---|---|---|---|---|---|---|
| | | | Scenario S1 | | | | | | | Scenario S2 | | | |
| CHE (units) | Gate 1 | Gate 2 | Gate 3 | Gate 4 | Gate 5 | Gate 6 | CHE (units) | Gate 1 | Gate 2 | Gate 3 | Gate 4 | Gate 5 | Gate 6 |
| RS | 8 | 9 | 6 | 3 | 3 | 6 | RS | 8 | 9 | 6 | 3 | 3 | 6 |
| TL | 12 | 5 | 2 | 4 | 1 | 6 | TL | 12 | 5 | 2 | 4 | 1 | 6 |
| YT | 14 | 8 | 5 | 4 | 2 | 8 | YT | 14 | 8 | 5 | 4 | 2 | 8 |
| Ratio | Rail:Road– 20:80 | | | | | | Ratio | Rail:Road– 50:50 | | | | | |

**Table 3. Settings for scenarios S3–S5 (the new LICD layout).**

| | New Layout | | |
|---|---|---|---|
| CHE (units) | Scenario 3 (S3) | Scenario 4 (S4) | Scenario 5 (S5) |
| RS | 18 | 24 | 30 |
| TL | 17 | 20 | 25 |
| YT | 30 | 30 | 35 |
| RTG | 0 | 1 | 1 |
| Ratio | | Rail:Road– 50:50 | |

**Table 4. Standard CHE usage rates.**

| CHE | Minimum (move/day) | | Maximum (move/day) | |
|---|---|---|---|---|
| | Move | Utilization | Move | Utilization |
| RS | 75 | 35% | 150 | 75% |
| TL | 40 | 20% | 150 | 75% |
| RTG | 60 | 35% | | |
| YT | 30 | 30% | | |

periods with great container demand. We therefore need to assess CHE usage rates—and thus the LICD's performance—during both peak and off-peak periods to ensure that the operations within the LICD are smoothed out.

## 4.2. Simulation model verification and validation

Prior to conducting a simulation, the proposed DES models have been initially assessed based on the results of simulation trials so that we could minimize the margin of error while ensuring that no desirable states, such as deadlocks, would be encountered during simulation runs. Nonetheless, only two DES models will be herein validated, namely the model that represents the current LICD operation (S1) and the one that represents the first phase of the new LICD (S3). It should also be remarked that, although the simulation period in this study is set as a whole year, the models are instead run in a quarter-wise fashion. This is due largely to the size of the models themselves that leads to an excessive amount of computational time in each experimental run. Moreover, all simulation results reported in this section are averaged over 30 replications.

Tables 5 and 6 below show discrepancies between the actual and the simulated container flows obtained from simulation trials for both settings. Regarding the current LICD layout, it could be seen from Table 5 that the simulated flows are relatively close to the actual data, since

**Table 5. Simulated container flows of the current LICD obtained from SIMIO (S1), together with the average margin of error in each quarter (in parentheses).**

| Quarter | Road | | Rail | |
|---|---|---|---|---|
| | Export (% Error[a]) | Import (% Error[a]) | Export (% Error[a]) | Import (% Error[a]) |
| Q1 | 152,088 (1.60%) | 118,323 (0.34%) | 39,068 (3.91%) | 30,682 (5.44%) |
| Q2 | 134,868 (1.27%) | 113,479 (0.10%) | 42,802 (3.05%) | 34,053 (4.09%) |
| Q3 | 125,073 (1.51%) | 109,653 (0.60%) | 39,406 (1.27%) | 31,459 (5.06%) |
| Q4 | 128,026 (1.58%) | 106,482 (0.22%) | 40,496 (2.49%) | 31,516 (6.83%) |
| Total | 540,056 | 447,938 | 161,772 | 127,710 |
| Average Error (%) | 1.49% | 0.31% | 1.45% | 4.74% |

[a]. % Error is computed from the average of errors $\left( \frac{|\text{actual flow} - \text{simulated flow}|}{\text{actual flow}} \cdot 100\% \right)$ over 30 replications.

the greatest average margin of error is only 4.74% at 0.90 confidence level. Likewise, the greatest average margin of error for the model representing the new LICD layout (Table 6) is also as low as 4.09% at 0.90 confidence level. Based on these numbers, it could be inferred that the performance of DES models is comparatively good, and these models could be used as representatives for the evaluation of both the current and the new LICD layouts under various scenarios.

## 4.3. Evaluating the current LICD layout

**4.3.1. The current LICD layout and operation (S1).** The performance of LICD, as measured by CHE utilization under the current operational setting (scenario S1) is provided in Table 7, where utilization rates of all CHE types at each gate operator during three different periods are reported. From Table 7, it could be seen that average CHE utilization of gate operator 3 is the greatest among all operators, and it is the only operator that properly runs cranes (RSs and TLs) according to the PAT's standard. Furthermore, YTs seem to be the least utilized among all CHE, as their usage rates are comparatively low and they do not vary greatly between peak and off-peak periods. The reason to this is due to the imbalance of container flow experienced by gate operators and the fact that there are currently too many privately owned yard trucks that ply within each gate operator's area. The situation is unfavorable as RSs are frequently deployed—in order to avoid double handling—via shifting operations, which further results in reduced YT usage rates.

**Table 6. Simulated container flows of the new LICD obtained from SIMIO (S3), together with the average margins of error for both import and export containers.**

| Quarter | Road | | Rail | |
|---|---|---|---|---|
| | Export | Import | Export | Import |
| Q1 | 91,430 | 84,675 | 80,156 | 63,572 |
| Q2 | 83,792 | 83,887 | 78,068 | 61,916 |
| Q3 | 76,202 | 81,441 | 79,112 | 62,744 |
| Q4 | 79,720 | 79,978 | 80,156 | 63,572 |
| Total | 331,144 | 324,981 | 317,492 | 251,804 |
| Ratio[a] | 53.54% | | 46.46% | |
| Average Export Error[b] | 3.21% | | | |
| Average Import Error[b] | 4.09% | | | |

[a]. Transportation ratio is computed based on the adjusted container flow and the current rail infrastructure, namely 64 TEUs per trip.

[b]. %Error is computed from the average of errors $\left( \frac{|\text{expected flow} - \text{simulated flow}|}{\text{expected flow}} \cdot 100\% \right)$ over 30 replications.

**Table 7. Simulation results for scenario S1.**

| Peak Period | | | | Off-peak Period | | | | Yearly Average | | | |
|---|---|---|---|---|---|---|---|---|---|---|---|
| Gate | RS | TL | YT | Gate | RS | TL | YT | Gate | RS | TL | YT |
| 1 | 30.75% | 4.52% | 3.76% | 1 | 20.41% | 3.20% | 3.08% | 1 | 28.78% | 3.73% | 3.37% |
| 2 | 28.14% | 11.90% | 4.79% | 2 | 20.76% | 4.59% | 4.37% | 2 | 23.39% | 7.65% | 4.60% |
| 3 | 50.92% | 45.03% | 11.95% | 3 | 45.14% | 35.30% | 9.51% | 3 | 49.00% | 41.04% | 11.15% |
| 4 | 24.43% | 13.50% | 2.72% | 4 | 19.59% | 5.83% | 2.22% | 4 | 21.27% | 9.06% | 2.46% |
| 5 | 45.12% | 22.07% | 4.56% | 5 | 32.44% | 17.26% | 3.50% | 5 | 37.31% | 18.82% | 3.80% |
| 6 | 49.88% | 22.11% | 9.07% | 6 | 37.38% | 12.71% | 6.60% | 6 | 45.18% | 18.29% | 7.96% |

**4.3.2. Scenario S1 and the new transportation policy (S2).** Scenario S2 differs from scenario S1 in terms of transportation setting—the transportation ratio between rail and road has changed from 20:80 to 50:50 according to the regulator's projection. To achieve this transportation ratio, some model parameters related to container flow need to be adjusted, including the number of containers arriving by trucks and trains, together with the frequency of cargo trains arriving at or leaving the LICD. Similar to scenario S1, the performance of LICD under scenario S2 is summarized and reported in Table 8.

Compared to scenario S1, the average utilization of some CHE types in scenario S2 has significantly improved, especially for the YTs (p-value < 0.001 at $\alpha = 0.10$), although such values are still far below the standard threshold set by the PAT, with an exception for gate operator 3. These results reveal that yard trucks are the most under-utilized assets among all CHE in both transportation settings, due largely to overcapacity and preference for shifting operations that have been described in the previous scenario.

In terms of crane utilization, the usage rates of RSs by some gate operators tend to be higher presumably because (*i*) reach stacker is the only CHE type capable of loading and unloading containers within rail areas and (*ii*) there are more train trips arriving to and departing from the LICD in this scenario.

## 4.4. Evaluating the new LICD layout

The new LICD layout will be evaluated at three different stages according to the PAT's multi-phase development plan—namely, scenarios S3, S4, and S5—each of which has different numbers of CHE, as previously summarized in Table 3. Instead of having six terminals operated by six independent gate operators, the new LICD comprises only two terminals, namely the north terminal and the south terminal, separated by railway tracks in the middle of its layout (see Fig 1b for additional details). All model parameters in these three scenarios are also adjusted in accordance with the new operational settings and sets of CHE deployed. For

**Table 8. Simulation results for scenario S2.**

| Peak Period | | | | Off-peak Period | | | | Yearly Average | | | |
|---|---|---|---|---|---|---|---|---|---|---|---|
| Gate | RS | TL | YT | Gate | RS | TL | YT | Gate | RS | TL | YT |
| 1 | 29.73% | 4.32% | 5.27% | 1 | 26.96% | 2.82% | 4.78% | 1 | 28.46% | 3.53% | 5.07% |
| 2 | 27.56% | 11.89% | 7.49% | 2 | 20.13% | 4.66% | 6.39% | 2 | 23.09% | 7.51% | 6.81% |
| 3 | 67.17% | 44.05% | 41.75% | 3 | 61.40% | 35.26% | 31.91% | 3 | 64.34% | 39.51% | 36.93% |
| 4 | 24.97% | 12.71% | 5.83% | 4 | 19.76% | 4.83% | 5.23% | 4 | 21.94% | 7.46% | 5.44% |
| 5 | 48.07% | 21.76% | 10.91% | 5 | 35.80% | 16.71% | 9.93% | 5 | 41.24% | 18.42% | 10.47% |
| 6 | 50.48% | 21.37% | 11.98% | 6 | 40.33% | 12.24% | 10.31% | 6 | 45.95% | 17.72% | 10.99% |

**Table 9. Simulation results for scenario S3.**

| North Terminal | | | | South Terminal | | | | LICD | | | |
|---|---|---|---|---|---|---|---|---|---|---|---|
| CHE | Peak Period | Off-peak Period | Yearly Average | CHE | Peak Period | Off-peak Period | Yearly Average | CHE | Peak Period | Off-peak Period | Yearly Average |
| RS | 71.35% | 51.63% | 68.27% | RS | 68.49% | 67.92% | 68.16% | RS | 69.50% | 67.51% | 68.20% |
| TL | 66.76% | 45.14% | 53.28% | TL | 39.07% | 37.12% | 38.04% | TL | 41.22% | 38.17% | 39.89% |
| YT | 22.80% | 18.25% | 19.67% | YT | 32.60% | 27.75% | 30.02% | YT | 29.15% | 24.40% | 26.37% |

instance, container flows at the upper and lower three terminals in the current LICD are combined and treated as container flows for the north and the south terminals of the new LICD, respectively. Moreover, the numbers of containers arriving by trucks and trains, as well as the frequency of train trips, in these scenarios are modified so that the resulting transportation ratio is as close as possible to 50:50.

**4.4.1. The initial phase of LICD's development plan (S3).** It could be seen from Table 3 that the numbers of CHE in the initial phase of LICD's development plan (S3) are relatively low in comparison to those of the current layout. The utilization rates of all CHE types in this scenario are therefore higher as reported in Table 9 —especially at the south terminal with relatively stable CHE utilization over the year, due to its larger operating area with more container handling activities.

Similar to the previous scenario, YTs seem to be the least utilized resources; however, this is due to different reasons. Particularly, in the current layout (scenarios S1 and S2), YTs are not well utilized mainly because of the imbalance between YTs and container demand experienced by gate operators that eventually leads to overcapacity issues, coupled with a preference for shifting operations. However, in this scenario, the performance of YTs is lessened mainly because of long waiting times induced by small fleets of cranes, as summarized in Table 10.

**4.4.2. The second phase of LICD's development plan (S4).** Scenario S4 differs from scenario S3 in terms of CHE variety and its respective numbers. In scenario S4, a rubber tyred gantry crane (RTG) is introduced into the rail zone for the improvement of rail transportation, along with the increasing numbers of cranes. Based on this CHE setting, gate-closing policy could also be modified, as operations in the north terminal are independent of those in the south one. Particularly, there is no need to clear all the external trucks circulating within the north terminal when a cargo train arrives on tracks adjacent to the south terminal, and vice versa.

The performance of LICD under scenario S4 could be evaluated as shown in Table 11. Similar to scenario S3, the results in Table 11 indicate that the south terminal has a superior performance in terms of CHE usage rates when compared to the north terminal. Although slightly above the PAT's standard, the utilization of YTs has exceeded this threshold for the first time, while that of RSs has been marginally lower than the upper limit set by the PAT, due largely to

**Table 10. Average waiting times per trip of yard trucks (YTs) in scenarios S1–S5 (hours).**

| Scenario | Peak Period | Off-peak Period | Yearly Average |
|---|---|---|---|
| S1 | 0.54 | 0.56 | 0.55 |
| S2 | 0.83 | 0.84 | 0.84 |
| S3 | 5.26 | 3.24 | 4.22 |
| S4 | 1.12 | 0.83 | 0.95 |
| S5 | 0.88 | 0.48 | 0.61 |

**Table 11. Simulation results for scenario S4.**

| North Terminal | | | | South Terminal | | | | LICD | | | |
|---|---|---|---|---|---|---|---|---|---|---|---|
| CHE | Peak Period | Off-peak Period | Yearly Average | CHE | Peak Period | Off-Peak Period | Yearly Average | CHE | Peak Period | Off-peak Period | Yearly Average |
| RS | 58.41% | 24.71% | 36.66% | RS | 60.42% | 50.82% | 55.60% | RS | 59.75% | 55.74% | 57.31% |
| TL | 58.05% | 56.01% | 57.28% | TL | 53.72% | 37.59% | 47.33% | TL | 48.50% | 43.30% | 46.10% |
| YT | 35.05% | 32.01% | 33.20% | YT | 37.98% | 30.27% | 33.74% | YT | 36.81% | 30.96% | 33.53% |
| - | - | - | - | - | - | - | - | RTG | 41.99% | 40.80% | 41.53% |

**Table 12. Simulation results for scenario S5.**

| North Terminal | | | | South Terminal | | | | LICD | | | |
|---|---|---|---|---|---|---|---|---|---|---|---|
| CHE | Peak Period | Off-peak Period | Yearly Average | CHE | Peak Period | Off-peak Period | Yearly Average | CHE | Peak Period | Off-peak Period | Yearly Average |
| RS | 44.54% | 42.49% | 43.64% | RS | 47.50% | 43.84% | 45.18% | RS | 46.51% | 43.54% | 44.66% |
| TL | 21.62% | 18.65% | 20.34% | TL | 34.20% | 30.24% | 32.40% | TL | 29.67% | 26.07% | 28.06% |
| YT | 26.33% | 25.29% | 25.74% | YT | 25.92% | 23.78% | 24.63% | YT | 26.08% | 24.38% | 25.08% |
| - | - | - | - | - | - | - | - | RTG | 42.33% | 40.54% | 41.35% |

the introduction of an RTG. To be precise, an RTG can perform the same functions as RSs. When it is used alongside YTs, an RTG can help reduce the workload of RSs while concurrently reducing the waiting time of YTs, which, in turn, leads to the rise and decline of YT and RS utilization rates, respectively.

**4.4.3. The final phase of LICD's development plan (S5).** Scenario S5 represents the LICD in a complete CHE setting, in which the numbers of RSs, TLs, and YTs are incrementally increased from those in scenario S4. The simulation results for scenario S5 are provided in Table 12 below. According to the table, the utilization rates of all CHE types significantly drop, as expected, because the numbers of CHE have increased but the container flow has remained constant.

In addition to CHE utilization, Table 13 shows the average time that each type of container spends at the LICD in all scenarios. From the table, it is evident that, in addition to low CHE usage rates, the existing LICD (scenario S1) is the poorest in terms of service times as average dwelling times for import and export containers are the highest. It is even inferior to scenario S3, which has fewer numbers of CHE. Although the dwelling times of scenarios S4 and S5 do not differ much from those of scenario S3, the ranges of such times tend to be lesser, implying smoother operations.

**Table 13. Average time that each container type spends at the LICD in scenarios S1–S5 (hours).**

| Peak Period | | | | Off-peak Period | | | | Yearly Average | | | |
|---|---|---|---|---|---|---|---|---|---|---|---|
| Scenario | Import | Export | Reefer | Scenario | Import | Export | Reefer | Scenario | Import | Export | Reefer |
| 1 | 75.86 | 56.90 | 43.47 | 1 | 65.52 | 49.14 | 35.65 | 1 | 71.66 | 53.74 | 40.97 |
| 2 | 55.01 | 41.25 | 42.39 | 2 | 53.73 | 40.30 | 41.84 | 2 | 55.15 | 40.37 | 42.20 |
| 3 | 72.13 | 60.11 | 44.90 | 2 | 46.50 | 38.75 | 33.85 | 3 | 51.82 | 43.19 | 44.81 |
| 4 | 56.29 | 46.91 | 42.03 | 3 | 53.46 | 44.55 | 34.12 | 4 | 55.79 | 46.49 | 41.11 |
| 5 | 53.89 | 44.91 | 38.72 | 4 | 53.69 | 44.74 | 34.17 | 5 | 53.79 | 44.84 | 35.30 |

## 4.5. The new LICD with incremental container flow (S6)

Based on scenarios S1 to S5, it could be inferred that the performance of LICD, as measured by CHE utilization, depends not only on the LICD layout but also on CHE, as well as the demand patterns experienced (*e.g.*, peak and off-peak periods). In terms of layout, the current LICD seems to underperform as the utilization rates of many CHE types are comparatively low when compared to the PAT's standard, especially for yard trucks. While we can improve the operational performance of LICD and thus the usage rates of expensive CHE through a new LICD layout (scenarios S3–S5), all of these results are, however, based on the current container flow. In order to gain insights into each of these scenarios, we have therefore assessed the LICD in a new scenario setting (scenario S6), in which container demands have been incrementally increased by 10% until they reach a level of 140% of the existing demand. With this piece of information, we would be able to determine the capability of LICD, as well as the points where the LICD should be upgraded so that CHE is properly utilized.

It should be remarked that, because LICD's capability could be defined by a period with peak container demand, scenario S6 will only be assessed based on the information of the first quarter, whose container flow is the greatest. With all these new settings, scenario S6 could be setup and simulated, whose results are summarized in Fig 10.

Fig 10a. represents CHE utilization under scenario S3 with incremental container demands—referred to as scenario S6-1. With a least-sized fleet of CHE, this LICD setting could accommodate up to 120% of the existing demand (approximately 1.51 million TEUs), before the utilization rate of RSs exceeds the maximum allowable usage rate. However, under such conditions, the utilization rates of other CHE types are still comparatively low, especially for YTs, as their utilization rate is barely above the PAT's standard (approximately 30.86%). This is mainly because of a long waiting time induced by the current gate-closing policy and limited number of RSs deployed.

Regarding the current gate-closing policy, when a cargo train arrives at a terminal (either the north or the south terminal), all terminal gates must be closed to prevent external trucks from entering the LICD. Full container loads will be then unloaded and loaded back onto a train with the help of RSs and YTs, while TLs need to wait and are rarely used for empty container handling until a train departs. Under a transportation ratio of 50:50 and a limited TL handling function, the utilization rate of TLs is unsurprisingly low.

Although YTs are complimentarily deployed alongside RSs for container handling activities in rail transportation, their usage rate is far lower than that of RSs as the numbers of RSs and YTs in scenario S6-1 are only 50% and 60% of those in the base case. YTs need to spend a majority of their time waiting for RSs, which eventually results in a low usage rate. This is even worse when the gates are open, as external trucks (and RSs) are allowed to perform similar operations as YTs, which further reduces the usage rate of YTs. To better utilize CHE under this LICD setting, we suggest that the gate-closing policy, along with train frequency, should be modified so that the uses of all CHE types are smoothed out.

When an RTG and some additional cranes are introduced to the LICD in scenario S6-2 (see Fig 10b.), the LICD is able to accommodate up to 140% of the existing demand (approximately 1.76 million TEUs). RS becomes the first CHE type that reaches the maximum allowable usage rate of 75%. The utilization rates of other CHE types—especially YTs—are also higher in this scenario due to reduced waiting times (see Table 10). We also observe that, when a level of 140% of the existing container demand is reached, almost all RSs and YTs will be operated within rail areas. As such, the LICD may not be able to accommodate container demand beyond this point.

Finally, Fig 10c. depicts CHE utilization of the LICD in scenario S6-3, which has full fleets of CHE. It could be seen that the usage rates of all CHE types lie well within the PAT's standard,

(a)

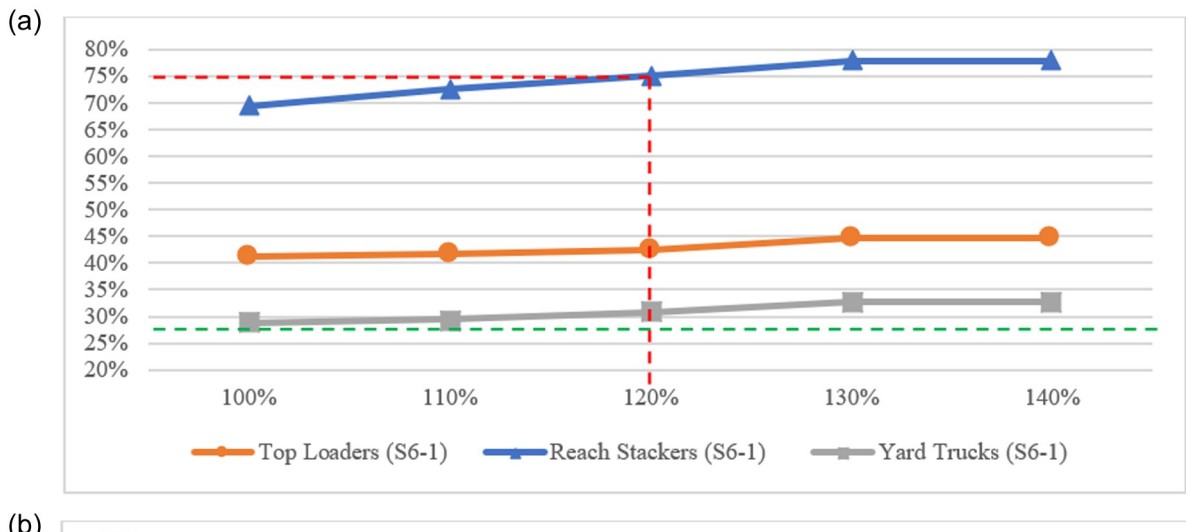

(b)

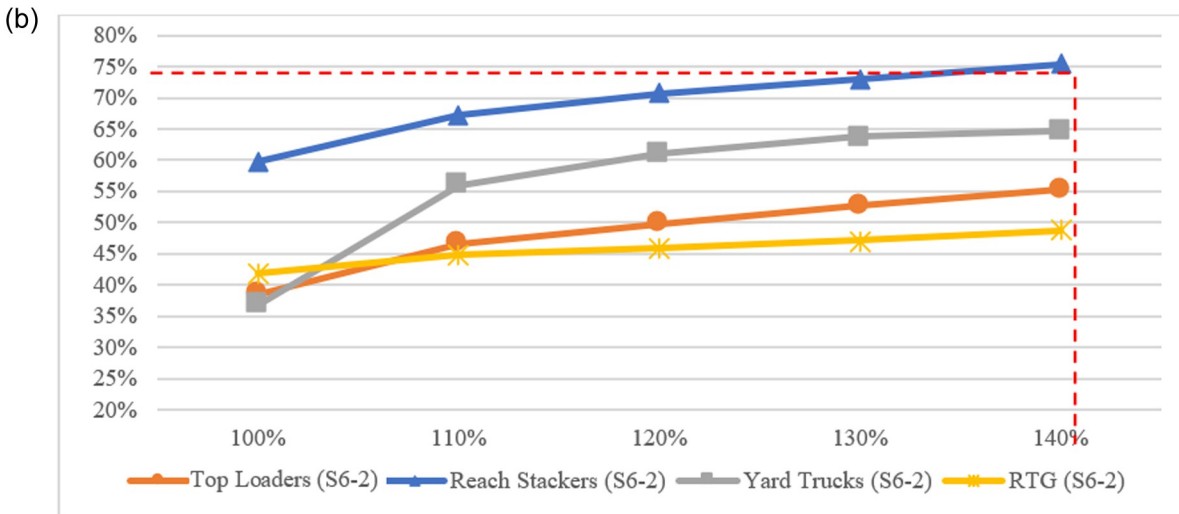

(c)

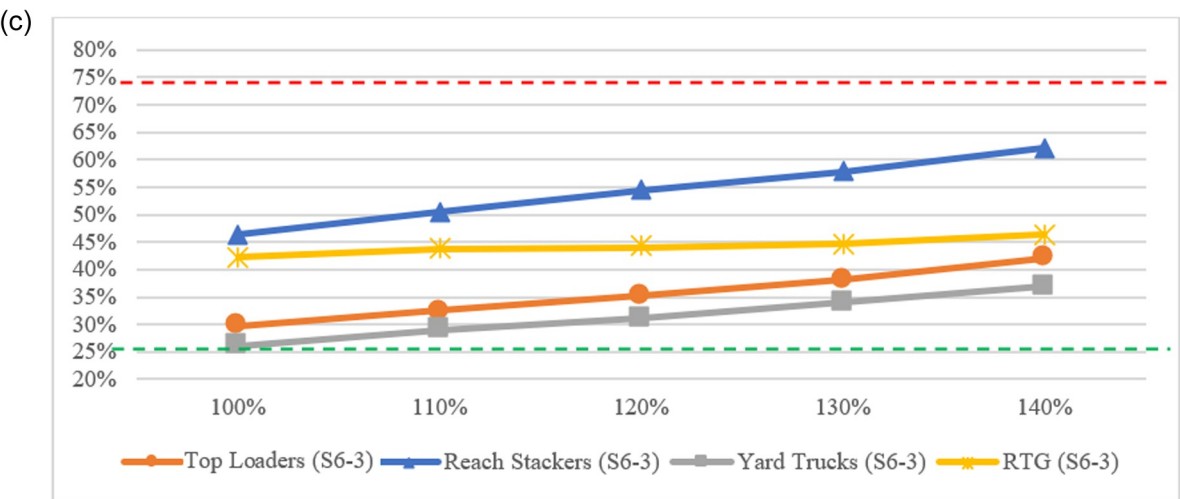

**Fig 10. CHE utilization of scenarios S6-1 to S6-3.** (a) CHE utilization in scenario S3 with incremental container demands (scenario S6-1). (b) CHE utilization in scenario S4 with incremental container demands (scenario S6-2). (c) CHE utilization in scenario S5 with incremental container demands (scenario S6-3).

even with a container demand of 140%—although YTs are not well utilized under some demand settings, while RSs are the most utilized CHE type as with previous scenarios. Based on these simulation results, we can conjecture that, when container demand rises further, the first type of CHE that would limit LICD's capability is the RS, followed by the TL. The reason for this is the saturation of container flow from rail transportation, where both capacity and frequency of trains could not be further increased. In such a situation, the additional container demand must move into the LICD via road networks, and the utilization rate of RSs would exponentially rise. However, the utilization rate of YTs would not abruptly rise, and may be unaffected, as YTs are currently used mostly in rail areas.

Also, this configuration may accommodate additional container flow. However, the imbalance in CHE usage might, unfortunately, deteriorate the LICD's performance. In order to enhance the performance of LICD, the Port Authority and the State Railway of Thailand should focus not only on the numbers of CHE but also on other operational issues, such as an increase in train capacity and a balanced train timetable, which could potentially enhance the long-term LICD's performance. All these issues could simply be assessed through the proposed DES models.

## 5. Discussion and conclusions

An inland terminal, or a dry port, is one of main facilities in multimodal transportation networks that helps expand the coverage of seaports while providing hinterland access to nearby seaports. While the impact of dry ports on the performance of multimodal transportation has become increasingly evident, research—especially that concerning a dry port's performance—is rather limited, as it involves not only specific factors related to the operations of dry ports themselves but also external factors, including changes in transportation policies and container demands.

In order to properly assess the operational performance of a dry port while taking into consideration both internal and external factors, a discrete event simulation (DES) framework has been herein developed using SIMIO simulation platform and applied to the Ladkrabang Inland Container Depot (LICD)—the first and only inland terminal in Thailand. Several DES models have been constructed to represent the LICD under six different scenarios: (*i*) the current LICD's operational setting (S1), (*ii*) the current LICD under a new transportation policy, where additional freight is moved through the LICD via rail transportation (S2), (*iii*) the initial phase of the new LICD with the least numbers of container handling equipment (CHE) (S3), (*iv*) the second phase of the new LICD with a rubber tyred gantry crane (RTG) (S5), (*v*) the final phase of the new LICD with full fleets of CHE, and (*iv*) the new LICD with incremental container demands (S6).

Our simulation results indicate that the operational performance of LICD is evidently affected by all the aforementioned factors, namely LICD layouts, CHE variety, and the numbers of CHE deployed, as well as the patterns of container demand experienced by the LICD. All in all, the current LICD setting (S1) has proven to be ineffective in terms of CHE utilization, as the usage rates of all CHE types are relatively low and varied across gate operators. Although we can boost the usage rates of CHE by moving additional containers via rail transportation (S2), it does not seem such a strategy would greatly improve the performance of LICD—especially for yard trucks (YTs)—due largely to the imbalance between operated CHE and container flows experienced by gate operators.

Our investigations also reveal that, by redesigning the LICD and its internal operations, the LICD's performance could be substantially enhanced even with limited numbers of CHE (S3). Furthermore, the LICD's performance seems to improve with the introduction of an

RTG (S4)—although the usage rates of all CHE types decline in scenario S5 due to excessive numbers of CHE.

In order to properly migrate from one CHE setting to another, the LICD is further assessed based on a scenario, in which container demands incrementally increase by 10% until they reach a level of 140% of the existing demand (S6). The simulation results of this scenario indicate that the capability of LICD in the early development stage is only 120% of the existing container demand (approximately 1.51 million TEUs), while that of the second phase is about 140%, or equivalently 1.76 million TEUs, before the usage rate of reach stackers (RSs) exceeds the maximum allowable rate of 75%. Moreover, the usage rate of YTs seems to positively correlate with the introduction of an RTG, as almost all of them are operated alongside an RTG within rail areas. Although the final phase of LICD performs well under a demand level of 140%, we conjecture that, if the demand further increases, the LICD would encounter operational difficulties induced by an excessive amount of container demand from road networks—as its rail transportation system is saturated in terms of both capacity and frequency. In such a case, the operational performance of LICD would be abruptly deteriorated from excessive RS movements. Notwithstanding such a fact, we can, fortunately, avoid such a circumstance by the proposed DES models, since the regulators are allowed to determine the best LICD configurations from the simulation conduct prior to the implementation.

Despite assumptions made to simplify the whole systems, we expect that our proposed DES models would be found useful to relevant players in the logistical industry, as the models could be applied in the performance evaluation of similar facilities. They could also be extended for use in the investigation of multimodal transportation networks with multiple types of facilities and modes of transportation—which is worth exploring in subsequent papers.

## Supporting information

**S1 Appendix. A list of abbreviations.**
(DOCX)

## Acknowledgments

The authors would like to thank Wattanasuk International Co., Ltd. and Mr.Wirattawut Boonbandansook for their help and support during the execution of this project. We would like to also thank the State Railway of Thailand (SRT) and Eastern Sea Laem Chabang Terminal Co., Ltd. (ESCO) for information and insights into the LICD operations, which are crucial for the completion of this research.

## Author Contributions

**Conceptualization:** Punyaanek Srisurin, Phipat Pimpanit, Pisit Jarumaneeroj.

**Formal analysis:** Punyaanek Srisurin, Phipat Pimpanit, Pisit Jarumaneeroj.

**Investigation:** Punyaanek Srisurin, Phipat Pimpanit, Pisit Jarumaneeroj.

**Methodology:** Phipat Pimpanit.

**Project administration:** Pisit Jarumaneeroj.

**Software:** Phipat Pimpanit.

**Supervision:** Pisit Jarumaneeroj.

**Validation:** Pisit Jarumaneeroj.

**Visualization:** Phipat Pimpanit.

**Writing – original draft:** Punyaanek Srisurin, Phipat Pimpanit.

**Writing – review & editing:** Pisit Jarumaneeroj.

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
