## [Decision Letter · Decision Letter 0]

22 Aug 2022

PONE-D-22-15457Evaluating long-term operational performance of a large-scale inland terminal: A discrete-event-simulation based modeling approachPLOS ONE

Dear Dr. Jarumaneeroj,

Thank you for submitting your manuscript to PLOS ONE. After careful consideration, we feel that it has merit but does not fully meet PLOS ONE’s publication criteria as it currently stands. Therefore, we invite you to submit a revised version of the manuscript that addresses the points raised during the review process.

We look forward to receiving your revised manuscript.

Kind regards,

Sakdirat Kaewunruen

Academic Editor

PLOS ONE

Journal Requirements:

Additional Editor Comments (if provided):

Please improve the quality of research as advised by the reviewers.

Reviewers' comments:

Reviewer's Responses to Questions

**Comments to the Author**

1. Is the manuscript technically sound, and do the data support the conclusions?

Reviewer #1: No

Reviewer #2: Yes

2. Has the statistical analysis been performed appropriately and rigorously? 

Reviewer #1: No

Reviewer #2: Yes

3. Have the authors made all data underlying the findings in their manuscript fully available?

Reviewer #1: No

Reviewer #2: Yes

4. Is the manuscript presented in an intelligible fashion and written in standard English?

Reviewer #1: No

Reviewer #2: Yes

5. Review Comments to the Author

Reviewer #1: The manuscript presented the discrete-event simulation to evaluate the long-term performance of the container terminal. Regretfully the manuscript needs to be rejected due to the following reasons:

1. Lack of novelty and contribution, the author only discusses the results from the Ladkrabang Inland Container Depot (LICD) case with no significant impact on the current development of container terminal efficiency for example automatization, double stack crane, etc.

2. The description of the methodology is unclear, no detailed algorithm has been explained, and the author only displays the DES model taken from the software used instead of writing the pseudocode.

3. All figures are presented in low quality and not in a good manner of high-impact publication.

4. There are no field measurements to validate the simulation results.

Reviewer #2: The study of the manuscript presents primary scientific research and propose decisions for increasing of container capacity in inland terminal. The analyses and amounts of the data presented are in sufficient details. For improvement of the paper, I have following remarks and suggestions:

1) The authors used too many abbreviations, such as CHE, LICD, SRT, BKR, PAT, YT, TL , ESCO, etc., that make the text difficult to read and especially for people that are not so close with this matter. Please try to reduce them.

2) In my opinion, in the abstract should be added some text about DES(Discrete Event Simulation) and why the authors choose it in their research as the only one appropriate method.

3) The list with references should be done in alphabetical order.

4) I recommend the authors to check once again the spelling and grammar in English before sending the revised manuscript. Some phrases (for clarity, unlike) and words (incremental) have to be replaced with appropriate synonyms.

6. PLOS authors have the option to publish the peer review history of their article (what does this mean?). If published, this will include your full peer review and any attached files.

Reviewer #1: No

Reviewer #2: No

---

## [Author Response · Author response to Decision Letter 0]

14 Sep 2022

Response to the reviewer's comments has been uploaded as a separate file.

---

## [Decision Letter · Decision Letter 1]

16 Nov 2022

PONE-D-22-15457R1Evaluating the long-term operational performance of a large-scale inland terminal: A discrete event simulation-based modeling approachPLOS ONE

Dear Dr. Jarumaneeroj,

Thank you for submitting your manuscript to PLOS ONE. After careful consideration, we feel that it has merit but does not fully meet PLOS ONE’s publication criteria as it currently stands. Therefore, we invite you to submit a revised version of the manuscript that addresses the points raised during the review process.

We look forward to receiving your revised manuscript.

Kind regards,

Sakdirat Kaewunruen

Academic Editor

PLOS ONE

Journal Requirements:

Reviewers' comments:

Reviewer's Responses to Questions

**Comments to the Author**

1. If the authors have adequately addressed your comments raised in a previous round of review and you feel that this manuscript is now acceptable for publication, you may indicate that here to bypass the “Comments to the Author” section, enter your conflict of interest statement in the “Confidential to Editor” section, and submit your "Accept" recommendation.

Reviewer #1: (No Response)

Reviewer #2: All comments have been addressed

Reviewer #3: (No Response)

2. Is the manuscript technically sound, and do the data support the conclusions?

Reviewer #1: No

Reviewer #2: Yes

Reviewer #3: Yes

3. Has the statistical analysis been performed appropriately and rigorously? 

Reviewer #1: No

Reviewer #2: Yes

Reviewer #3: N/A

4. Have the authors made all data underlying the findings in their manuscript fully available?

Reviewer #1: No

Reviewer #2: Yes

Reviewer #3: No

5. Is the manuscript presented in an intelligible fashion and written in standard English?

Reviewer #1: No

Reviewer #2: Yes

Reviewer #3: Yes

6. Review Comments to the Author

Reviewer #1: No significant improvement is observed. For example 1) the contribution of the paper is merely local cases of LICD; 2) the description of the methodology is still unclear; 3) all figure is low quality and lack an important point of the figure.

We suggest submitting to another more suitable journal rather than Plos One.

Reviewer #2: (No Response)

Reviewer #3: The abstract should be improved and must be cleared, especially the result. I would also suggest the authors to reduce number of abbreviation along the manuscript.

7. PLOS authors have the option to publish the peer review history of their article (what does this mean?). If published, this will include your full peer review and any attached files.

Reviewer #1: No

Reviewer #2: No

Reviewer #3: No

---

## [Author Response · Author response to Decision Letter 1]

20 Nov 2022

Responses have been uploaded separately.

---

## [Editor Report · Decision Letter 2]

22 Nov 2022

Evaluating the long-term operational performance of a large-scale inland terminal: A discrete event simulation-based modeling approach

PONE-D-22-15457R2

Dear Dr. Jarumaneeroj,

We’re pleased to inform you that your manuscript has been judged scientifically suitable for publication and will be formally accepted for publication once it meets all outstanding technical requirements.

Kind regards,

Sakdirat Kaewunruen

Academic Editor

PLOS ONE

Additional Editor Comments (optional):

Generally the technical content of the manuscript is fine. However, please improve the quality of presentations and editorial matters. For example, the graphs plotted and obtained directly from Excel are of poor quality. The authors should consider replot all of the graphs using professional data visualisation software. In addition, most figures are of poor quality. Some figures cannot be read properly. Please improve these before submitting the final version.
---

## [Editor Report · Acceptance letter]

23 Nov 2022

PONE-D-22-15457R2 

Evaluating the long-term operational performance of a large-scale inland terminal: A discrete event simulation-based modeling approach 

Dear Dr. Jarumaneeroj:

I'm pleased to inform you that your manuscript has been deemed suitable for publication in PLOS ONE. Congratulations! Your manuscript is now with our production department. 

Kind regards, 

on behalf of

Dr. Sakdirat Kaewunruen 

Academic Editor

PLOS ONE